# Maturation of a central brain flight circuit in *Drosophila* requires Fz2/Ca²⁺ signaling

**Tarjani Agrawal, Gaiti Hasan***

National Centre for Biological Sciences, Tata Institute of Fundamental Research, Bangalore, India

**Abstract** The final identity of a differentiated neuron is determined by multiple signaling events, including activity dependent calcium transients. Non-canonical Frizzled2 (Fz2) signaling generates calcium transients that determine neuronal polarity, neuronal migration, and synapse assembly in the developing vertebrate brain. Here, we demonstrate a requirement for Fz2/Ca²⁺ signaling in determining the final differentiated state of a set of central brain dopaminergic neurons in *Drosophila*, referred to as the protocerebral anterior medial (PAM) cluster. Knockdown or inhibition of Fz2/Ca²⁺ signaling during maturation of the flight circuit in pupae reduces *Tyrosine Hydroxylase* (*TH*) expression in the PAM neurons and affects maintenance of flight. Thus, we demonstrate that Fz2/Ca²⁺ transients during development serve as a pre-requisite for normal adult behavior. Our results support a neural mechanism where PAM neuron send projections to the α' and β' lobes of a higher brain centre, the mushroom body, and function in dopaminergic re-inforcement of flight.

## Introduction

Genetically encoded developmental programs and neuronal activity together shape the neurotransmitter identity of developing neural circuits. In vertebrates, calcium transients generated by neuronal activity can influence neurotransmitter specification during development and in adults (*Spitzer, 2012*; *Borodinsky et al., 2014*). One mechanism of generating Ca²⁺ transients is non-canonical Wnt/Ca²⁺ signaling initiated by membrane bound Frizzled receptors and a trimeric G-protein (*Slusarski et al., 1997*). Such calcium signals are known to affect neuronal polarity, migration as well as synapse assembly in the developing and mature vertebrate brain (*Varela-Nallar et al., 2010*; *Ciani et al., 2011*). Wnt signaling was first identified in *Drosophila* where multiple genes encode Wnt and Fz proteins (*van Amerongen and Nusse, 2009*). However, the role of non-canonical Wnt/Ca²⁺ signaling during neural development and circuit maturation is poorly understood in invertebrates and its ability to stimulate Ca²⁺ transients during circuit maturation is unknown. In a screen for G-protein coupled receptors required for flight circuit maturation in *Drosophila* we identified *dFrizzled2* (*dFz2*) and found that flight deficits upon *dFz2* knockdown can be suppressed by over-expression of the intracellular endoplasmic reticular Ca²⁺ sensor dSTIM (*Agrawal et al., 2013*). Adult neural circuits in *Drosophila*, including the flight circuit, form in the pupal stages (*Consoulas et al., 2002*), and it is known that maturation of the *Drosophila* flight circuit requires intracellular Ca²⁺ signaling (*Banerjee et al., 2004*; *Agrawal et al., 2013*). To understand the molecular and cellular basis for such flight deficits, we set out to map neurons that require dFz2 receptor signaling in the context of flight circuit maturation.

Insect flight requires computation of multiple sensory inputs and their integration with the flight motor system. This computation and integration presumably occurs in central neurons and allows for control of initiation, maintenance and cessation of voluntary flight bouts (*Gotz, 1987*; *Strauss, 2002*; *Strausfeld and Hirth, 2013*). Recent work has shown that central dopaminergic neurons in the ventral ganglion modulate a pair of direct flight muscle motor neurons required for wing coordination during flight initiation and cessation (*Sadaf et al., 2015*). In addition, central neurons that compute sensory

*For correspondence: gaiti@ncbs.res.in

Competing interests: The authors declare that no competing interests exist.

**eLife digest** The fruit fly *Drosophila melanogaster* is an aerial acrobat. These insects can suddenly change direction in less than one hundredth of a second, explaining why a moving fly can be so difficult to swat. To perform their aerial manoeuvres, the flies continually combine information from multiple senses, including vision, hearing and smell, and use these data to control the activity of the neural circuits that support flight.

These flight circuits are established during the pupal stage of fly development, during which the fly transforms from a larva into its adult form. In 2013, researchers showed that a protein called dFrizzled2 must be present in pupae for flight circuits to mature correctly. This protein forms part of a pathway that ultimately controls which specific chemicals—called neurotransmitters—are released by neurons to communicate with other cells. Agrawal and Hasan—who worked on the 2013 study—now extend their findings to investigate the role of dFrizzled2 in more detail.

The new experiments show that for the flight circuits to mature, dFrizzled2 must be active in a cluster of neurons known collectively as PAM. Specifically, dFrizzled2 is needed to make an enzyme that helps to produce a neurotransmitter called dopamine. This in turn enables the PAM neurons to communicate with a region of the fruit fly brain called the mushroom body, which it thought to play an important role in complex behaviors such as reward-based learning.

The absence of dFrizzled2 results in adult flies that rarely remain airborne for more than 20 s at a time, whereas normal flies can typically fly for over 700 s. Given that dopamine is known to signal reward, one possibility is that the dopamine signals from the PAM neurons to the mushroom body serve as a reward to encourage continuous flight. Mutant flies that lack dFrizzled2—and thus these dopamine signals—lose their motivation to fly after only a few seconds.

Overall, Agrawal and Hasan's findings suggest that the mushroom body has an important role in coordinating a fly's movements with information from it senses. Future research will be needed to determine exactly how the mushroom body performs this role.

information in real time and control the timing of a flight bout must exist but remain unknown. Most complex insect behaviors, including flight, are modulated by various monoamines and neuropeptides and in *Drosophila*, flight can be modulated by octopamine, serotonin and dopamine as well as several neuropeptides (*Taghert and Nitabach, 2012*; *Sadaf and Hasan, 2014*; *Van Breugel et al., 2014*). Here, we show for the first time that dFz2 signaling drives the expression of *Tyrosine Hydroxylase* (TH), the rate-limiting enzyme in dopamine synthesis (*Friggi-Grelin et al., 2003*), during circuit maturation, in a specific set of central brain dopaminergic neurons, called the protocerebral anterior medial (PAM) neurons. The PAM cluster consists of approximately 90 dopaminergic neurons, which project to different regions of a higher brain structure called the mushroom body (MB). PAM-MB connectivity has been studied for its role in olfactory associative learning and memory (*Aso et al., 2012*; *Liu et al., 2012*) where it is thought to signal reward reinforcement. More recently, a PAM-MB circuit was shown to control negative geotaxis behavior in flies (*Riemensperger et al., 2013*). Our studies demonstrate the presence of a novel PAM-MB flight circuit and support a role for PAM-MB synapses in dopaminergic re-inforcement of flight bouts.

## Results

### Flight deficits in adult *Drosophila* arise from reduced dFz2 function in pupal dopaminergic neurons

To identify neurons, which require dFz2 function for flight, an RNAi strain (*dFz2-IR*) was expressed in independent neurotransmitter domains with the help of the *UASGAL4* system for cell and tissue specific expression (*Brand and Perrimon, 1993*). Amongst the neuronal domains tested, significant flight deficits were observed upon knockdown of *dFz2* in aminergic neurons (60% flight time; *DdcGAL4*) and in dopaminergic neurons (45% flight time; *THGAL4)* (*Figure 1A*, *Video 1*). Because, *DdcGAL4* drives expression in serotonergic and dopaminergic neurons, we tested flies with knockdown of *dFz2* in serotonergic neurons (*TRHGAL4*). These flies exhibit normal flight bouts in the tethered flight assay (*Figure 1A*). Similarly, normal flight bouts were observed in flies with

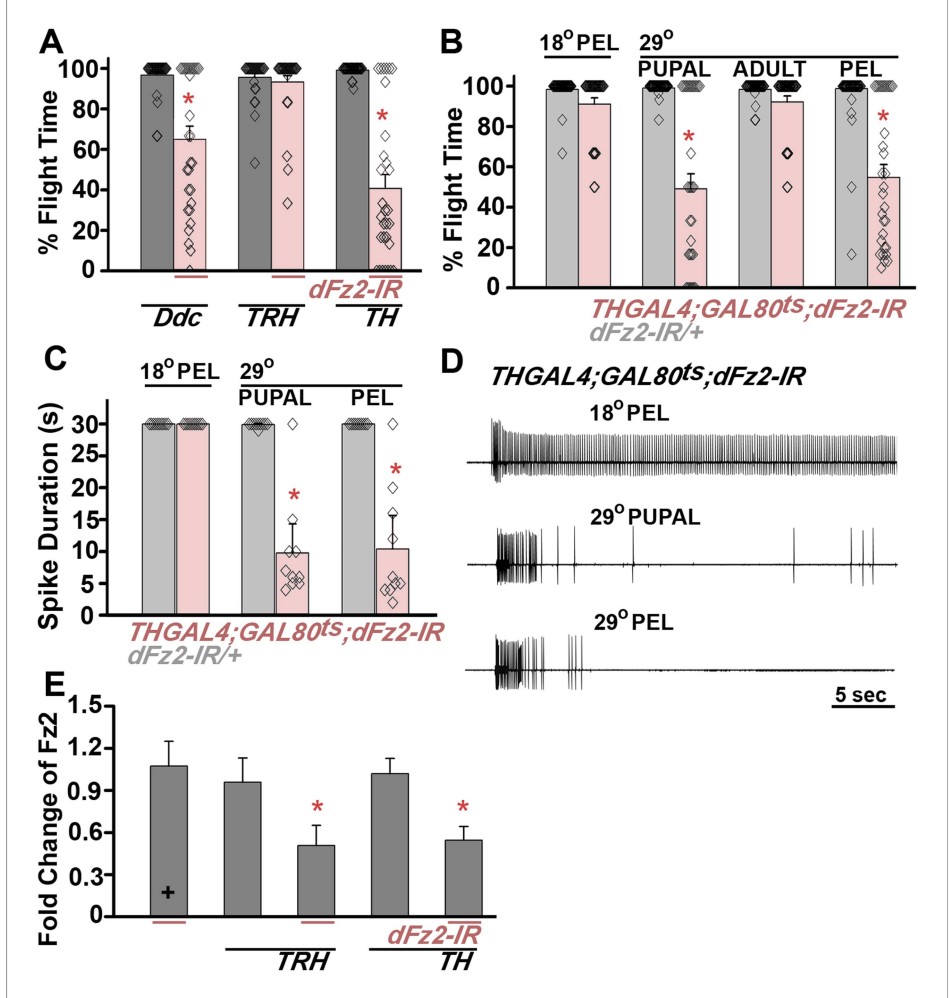

**Figure 1**. dFz2 function is required in dopaminergic neurons during development for normal adult flight. (**A**) Percentage flight times of individuals after knockdown of *dFz2* in aminergic neurons (*DdcGAL4*), serotonergic neurons (*TRHGAL4*), dopaminergic neurons (*THGAL4*). Knockdown of *dFz2* in aminergic neurons (*DdcGAL4*, first bar in red) and in dopaminergic neurons (*THGAL4*, third bar in red) showed reduced flight. Knockdowns were compared to their respective GAL4 controls (gray bars; *p < 0.001, Mann–Whitney U-test). (**B**) Percentage flight times of *dFz2-IR* heterozygotes (gray bars) and flies with knockdown of *dFz2* in dopaminergic neurons (red bars) at specific developmental stages by temperature controlled *THGAL4; GAL80ts* expression are shown. Flies with knockdown during pupal development exhibit reduced flight similar to knockdown post-egg laying (PEL) as compared to controls (*p < 0.001, Mann–Whitney U-test). (**C**) Durations of rhythmic action potentials recorded from the DLMs of air-puff stimulated tethered flies. Bars represent the mean spike duration and diamonds represent the spike duration of an individual recording (*p < 0.001, Mann–Whitney U-test). (**D**) Representative traces of electrophysiological recordings from DLMs of individuals with *dFz2* knockdown at the indicated developmental stages are shown. (**E**) Quantification of *dFz2* transcript levels after knockdown by *dFz2* RNAi in serotonergic (*TRHGAL4*) and dopaminergic (*THGAL4*) neurons. The Y-axis represents log2 fold changes calculated by the ΔΔCt method. Each value is the mean ± SEM of three independent experiments, obtained from three independent RNA samples (*p < 0.05, one-way ANOVA).

The following figure supplements are available for figure 1:

**Figure supplement 1**. Normal flight in flies with knockdown of dFz2 in non-dopaminergic neurons.

**Figure supplement 2**. Expression of multiple dFz2-IR strains in dopaminergic neurons exhibits flight defects.

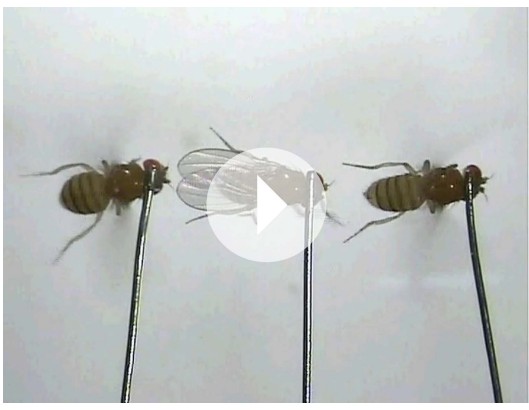

**Video 1.** dFz2 knockdown in dopaminergic neurons result in flight defect. Real time video recording of air-puff induced flight in the following genotypes from left to right. (1) *THGAL4/+*, (2) *THGAL4;dFz2-IR*, (3) *dFz2-IR/+*. Following a gentle air-puff *THGAL4;dFz2-IR* flies were able to initiate but not maintain flight for as long as control flies of the genotypes *THGAL4/+* and *dFz2-IR/+*. DOI: 10.7554/eLife.07046.006

knockdown of *dFz2* by *OK371GAL4* (mainly glutamatergic neurons), *GADGAL4* (many GABA-ergic neurons), and *P386GAL4* (a peptidergic neuron subset that expresses in cells with the neuropeptide processing enzyme *amontillado*; *Figure 1—figure supplement 1*). Thus, dFz2 function is required primarily in flight circuit neurons, which express GAL4 under control of the *pale* (*ple*) gene promoter that codes for TH. TH catalyzes a rate-limiting enzymatic step in the synthesis of dopamine (*Friggi-Grelin et al., 2003*). Therefore, expression of this enzyme is considered a reliable marker of dopaminergic neurons. The *THGAL4* strain marks a high proportion of TH-positive neurons in the *Drosophila* central nervous system (*Friggi-Grelin et al., 2003*).

An earlier study demonstrated that pan-neuronal knockdown of *dFz2* in pupae leads to flight deficits. However, *dFz2* knockdown in adults did not affect flight (*Agrawal et al., 2013*). Existing larval progenitor neurons undergo extensive re-modeling of their axonal and dendritic arbors during pupal stages to form synaptic connections of the mature adult flight circuit (*Fernandes and VijayRaghavan, 1993*; *Consoulas et al., 2002*). We therefore determined the developmental stage at which dFz2 function is required in dopaminergic neurons for flight. For this purpose, we used the TARGET (temporal and regional gene expression targeting) system (*McGuire et al., 2003*). TARGET regulates GAL4 expression by a temperature sensitive GAL80ts element, which can be expressed and repressed at 18°C and 30°C, respectively (*McGuire et al., 2004*). Experimental animals of the genotype *THGAL4, GAL80ts;dFz2-IR* were shifted to the permissive GAL4 expression temperature (30°C) either during pupal or adult stages. This allowed stage-specific knockdown of *dFz2*. Upon knockdown of *dFz2* in *TH*-expressing neurons through pupal development a significant reduction of flight time was observed, whereas normal flight bouts, as measured for 30 s, were observed upon knockdown of *dFz2* in adults (*Figure 1B*). Flight deficits upon knockdown of *dFz2* in pupae were equivalent to those observed upon knockdown throughout post-embryonic development confirming that dFz2 requirement for flight is primarily in pupal dopaminergic neurons during circuit maturation. A physiological correlate of flight is rhythmic patterns of action potentials recorded from the dorsal longitudinal muscles (DLMs) during tethered flight. A reduction in duration of flight patterns was observed upon knockdown of *dFz2* in *TH*-expressing neurons during pupal development (*Figure 1C,D*). *dFz2* knockdown at pupal and adult stages in *TH*-expressing neurons was confirmed by qPCR. As a control, we also confirmed *dFz2* knockdown in serotonergic neurons targeted by *TRHGAL4* where no flight deficits were observed (*Figure 1E*). The specificity of flight deficits obtained upon *dFz2* knockdown was tested by expression of three additional RNAi strains for *dFz2* (*BL27568, BL31390*, and *BL31312*). Significant flight defects ranging from 67% to 72% were obtained upon knockdown through post-embryonic development (*Figure 1—figure supplement 2*). The difference in flight deficits between *dFz2-IR* and the three other RNAi strains is very likely due to a difference in their efficacy of knockdown (compare *Figure 1E* with *Figure 1—figure supplement 2B*). Therefore, the *dFz2-IR* strain was used for all subsequent analyses.

## dFz2 is required in the PAM dopaminergic neurons for flight

Dopaminergic neurons marked by *THGAL4* have been broadly classified into seven clusters in the brain (*Figure 2A*; *Table 1*). In addition *TH*-expressing neurons are present in each segment of the ventral ganglion (*Mao and Davis, 2009*; *Sadaf et al., 2015*). In the brain, two neuronal clusters referred to as PAM and PAL (Protocerebral Anterior Lateral) are located in the anterior region, whereas five neuronal clusters, PPM1, 2, and 3 (Protocerebral Posterior Medial), PPL1 and 2 (Protocerebral Posterior Lateral) are located in the posterior region (*Figure 2A*). In order to identify

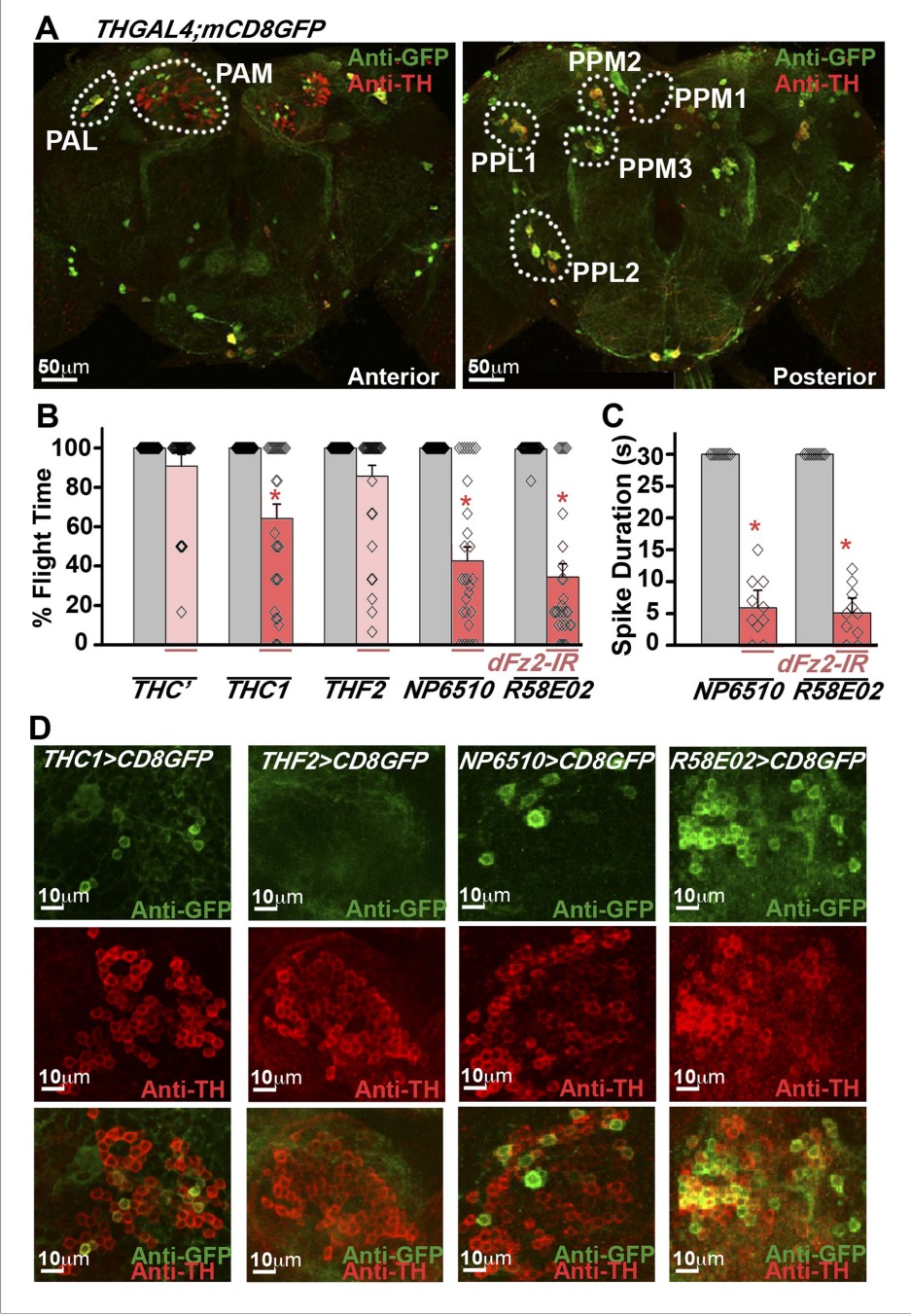

**Figure 2**. RNAi-mediated knockdown of dFz2 function in Protocerebral Anterior Medial (PAM) dopaminergic neurons causes flight deficits. (**A**) Expression pattern of *THGAL4* in the anterior and posterior regions of the brain are shown. Dotted line markings show the neuronal clusters. PAM: protocerebral anterior medial; PAL: protocerebral anterior lateral; PPM1, PPM2, PPM3: protocerebral posterior medial 1, 2, and 3; PPL1, PPL2: protocerebral posterior lateral 1 and 2. (**B**) Percentage flight times of heterozygous GAL4 controls (gray bars) and GAL4-specific knockdown of *dFz2* (red bars). Knockdown of *dFz2* in PAM-expressing GAL4 individuals (*THC1GAL4, NP6510GAL4, R58E02GAL4*) resulted in significantly reduced flight times when compared to their respective GAL4 controls (*p < 0.001, Mann–Whitney U-test). (**C**) Durations of rhythmic action potentials recorded from the DLMs of air-puff stimulated tethered flies. Average Spike durations were reduced upon expression of *dFz2* RNAi in *NP6510GAL4* and *R58E02GAL4* as compared to GAL4s controls (*p < 0.001, Mann–Whitney U-test). (**D**) Expression of *THC1GAL4, THF2GAL4, NP6510GAL4*, and *R58E02GAL4* in the PAM neuronal cluster is shown. Except

*Figure 2. continued on next page*

*Figure 2. Continued*

*THF2GAL4*, all other GAL4s express in dopaminergic PAM neurons. Expression was analyzed from 10 brain hemispheres.

The following figure supplements are available for figure 2:

**Figure supplement 1**. Electrophysiological traces showed reduced firing upon knockdown of *dFz2*.

**Figure supplement 2**. Expression of dFz2 in PAM dopaminergic neurons.

**Figure supplement 3**. Expression of TH in PAM dopaminergic neurons during development.

*TH*-expressing neurons that require dFz2 function for flight, three independent GAL4 strains (*THC'*, *THC1*, and *THF2*; *Liu et al., 2012*), with differential expression in central brain clusters and the ventral ganglion, were tested (*Table 1*). Significant flight deficits were observed upon expression of *dFz2-IR* under control of *THC1GAL4,* but not with *THC'GAL4* and *THF2GAL4* (*Figure 2B*). These data suggested that either all or some neurons in the PAM, PPM1, and T3 regions, marked by *THC1GAL4,* but poorly marked or not marked by *THC'GAL4* and *THF2GAL4,* form part of the flight circuit and require dFz2 signaling during pupal development. Next, we tested two strains (*NP6510GAL4* and *R58E02GAL4*) (*Riemensperger et al., 2013*) which drive expression uniquely in the PAM neurons (*Table 1*). Significant flight deficits were observed in flies with knockdown of *dFz2* by either *NP6510GAL4* or *R58E02GAL4* (*Figure 2B,C*, *Figure 2—figure supplement 1*, *Video 2*) implicating these dopaminergic neurons as part of a central brain flight circuit. These data do not rule out a role for additional central neurons or ventral ganglion neurons in the regulation of flight. Expression of dFz2 was confirmed in adult PAM neurons by immunohistochemistry (*Figure 2—figure supplement 2*). Knockdown by *dFz2-IR* in PAM neurons resulted in significant loss of dFz2 immunostaining (*Figure 2—figure supplement 2*). Moreover, in support of the pupal requirement for dFz2 (*Agrawal et al., 2013*) (*Figure 1B*), PAM neurons marked by *R58E02GAL4* do not express TH in the larval stages (*Figure 2—figure supplement 3*), indicating that TH immunoreactivity in these neurons is acquired during pupal maturation, as observed by co-localization of TH immunostaining with *R58E02GAL4*-driven GFP in pupae (*Figure 2—figure supplement 3*). With this we concluded that PAM neurons require dFz2 signaling during functional maturation of the flight circuit in pupae.

**Table 1**. Summary of expression pattern of GAL4s

|  | *THC'* | *THC1* | *THF2* | *NP6510* | *R58E02* |
|---|---|---|---|---|---|
| PAM | + | ++ | − | ++ | +++ |
| PAL | + | + | − | − | − |
| PPM1 | + | ++ | − | − | − |
| PPM2 | + | + | + | − | − |
| PPM3 | − | − | + | − | − |
| PPL1 | − | + | ++ | − | − |
| PPL2 | + | + | + | − | − |
| T1 | + | + | − | − | − |
| T2 | − | + | + | − | − |
| T3 | − | + | − | − | − |
| Ab | + | + | + | − | − |

Table summarizing the expression pattern of *THC'GAL4, THC1GAL4, THF2GAL4, NP6510GAL4*, and *R58E02GAL4* in specified dopaminergic neuronal clusters. Clusters shown in **Figure 2A** and thoracic ganglion (T1, T2, T3, Ab) were examined for the expression. Plus (+) and minus (−) indicate the presence and absence of dopaminergic positive neurons, respectively. Double plus (++) and triple plus (+++) indicate the presence of >5 and >50 dopaminergic positive neurons, respectively. 10 brain hemispheres were analyzed for the expression.

## dFz2 regulates flight circuit development through non-canonical Fz2/Ca²⁺ signaling

A role for dFz2 in flight circuit development was originally identified in a screen for GPCRs that signal through changes in intracellular $Ca^{2+}$ (*Agrawal et al., 2013*). In order to identify molecules that function downstream of Fz2 for development of the adult flight circuit, interactions of candidate genes were tested. Reports from vertebrates suggest that dFz2 activates downstream $Ca^{2+}$ signaling through non-canonical

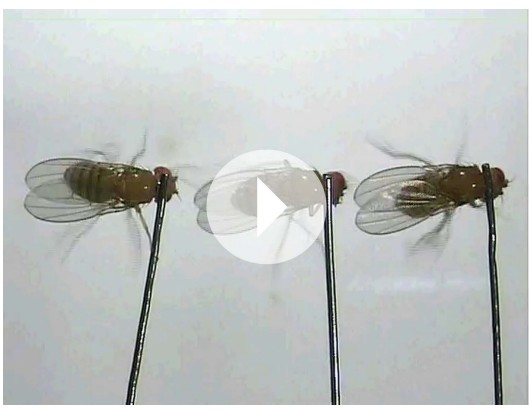

**Video 2.** dFz2 knockdown in PAM neurons result in flight defect. Real time video recording of air-puff induced flight in the following genotypes from left to right. (1) *R58E02GAL4/+*, (2) *R58E02GAL4;dFz2-IR*, (3) *dFz2-IR/+*. Following a gentle air-puff *R58E02GAL4; dFz2-IR* flies were able to initiate but not maintain flight for as long as control flies of the genotypes *THGAL4/+* and *dFz2-IR/+*.

mechanisms (*Slusarski et al., 1997*). We tested flight in animals with *dFz2* knockdown in combination with both canonical (*Figure 3—figure supplement 1*) and non-canonical (*Figure 3A*) candidate genes. In the canonical signaling pathway dFz2 along with its co-receptor lipoprotein receptor related-protein 5/6 (LRP5/6, encoded by *Arrow* in *Drosophila*) activates Dishevelled. Activated dishevelled functions to stabilize β catenin (*Drosophila armadillo*) and hence promote β catenin entry into the nucleus followed by enhanced transcription of downstream target genes. We over-expressed wild-type Dishevelled, a point mutant (G64V) in the DIX-domain of Dishevelled (specifically activates the canonical pathway; *Penton et al., 2002*) and a constitutively active form of Armadillo (*UAS-armS10*; *Morel and Arias, 2004*) in the background of *dFz2* down-regulation either in dopaminergic neurons or across all neurons. The resultant progeny were tested for flight. Up-regulation of canonical signaling molecules did not rescue flight deficits of *dFz2* down-regulation (*Figure 3—figure supplement 1*). Moreover, normal flight times were observed in flies with RNAi knockdown of canonical dFz2 pathway components like LRP5/6, Dishevelled, and GSK3β in either the pan-neuronal domain or in dopaminergic neurons (*Figure 3—figure supplement 1*). RNAi strains for dishevelled and GSK3β were validated by quantitative PCR (qPCR; *Figure 3—figure supplement 1*), whereas RNAi for LRP5/6 was validated by ubiquitous expression with *Act5CGAL4* that resulted in embryonic lethality (*Dietzl et al., 2007*). These results do not support a role for canonical dFz2 signaling in dopaminergic neurons for maturation of the flight circuit in pupae.

Next, we tested flight deficits by expression of previously implicated non-canonical candidates that link dFz2 activation with Ca²⁺ signaling (*Figure 3A*) (*Sheldahl et al., 2003*). From genetic studies, we know that the heterotrimeric G-protein, Gq, which links GPCR activation to intracellular store calcium release, does not function downstream of dFz2 signaling in the context of *Drosophila* flight (*Agrawal et al., 2013*). Therefore, we tested the requirement of other heterotrimeric G-proteins from *Drosophila*. Constitutively, active forms of the α subunits of Gs (*UASAcGs*), Go (*UASAcGo*), and Gi (*UASAcGi*) were tested in flies with *dFz2* knockdown in pupal stages. Pan-neuronal expression of AcGo in pupae with *dFz2* knockdown, rescued flight defects to a significant extent (*Figure 3B*), whereas constitutively active forms of Gi, Gq, or Gs did not (*Figure 3—figure supplement 2*). A partial rescue of flight defects was also observed upon AcGo expression in dopaminergic neurons in pupae (*Figure 3B*). These data support Go activation by dFz2 in dopaminergic neurons of the maturing flight circuit during pupal development.

To confirm the requirement of Go in dopaminergic neurons, we down-regulated Go function either by expression of an RNAi construct (*Vecsey et al., 2014*) *THGAL4;GoRNAi* or by expression of pertussis toxin which inhibits Go function (*THGAL4;UASPTX.16*) and tested the progeny for flight. In *Drosophila*, pertussis toxin is a selective inhibitor of Go signaling (*Hopkins et al., 1988*; *Ferris et al., 2006*). Expression of *Go-IR* and *PTX.16* in dopaminergic neurons reduced both flight times (*Figure 3B*), and the maintenance time of flight patterns recorded from the DLMs (*Figure 3D,E*). Moreover, down-regulation of Go signaling in PAM neurons with *R58E02GAL4*, by expression of either *Go-IR* or *PTX.16* resulted in significant loss of flight (*Figure 3B*). PTX induced flight deficits required expression in pupae and not in adults (*Figure 3B* and *Figure 3—figure supplement 2*).

Flight deficits induced by pan-neuronal knockdown of *dFz2* were rescued to a significant extent by over-expression of the ER Ca²⁺ depletion sensor, *dSTIM⁺* (*Agrawal et al., 2013*), suggesting that activation of Go by dFz2 evokes Ca²⁺ signals in *Drosophila* neurons. As in other organisms (*Feske et al., 2006*; *Prakriya et al., 2006*; *Vig et al., 2006*), in *Drosophila* neurons as well Ca²⁺ release

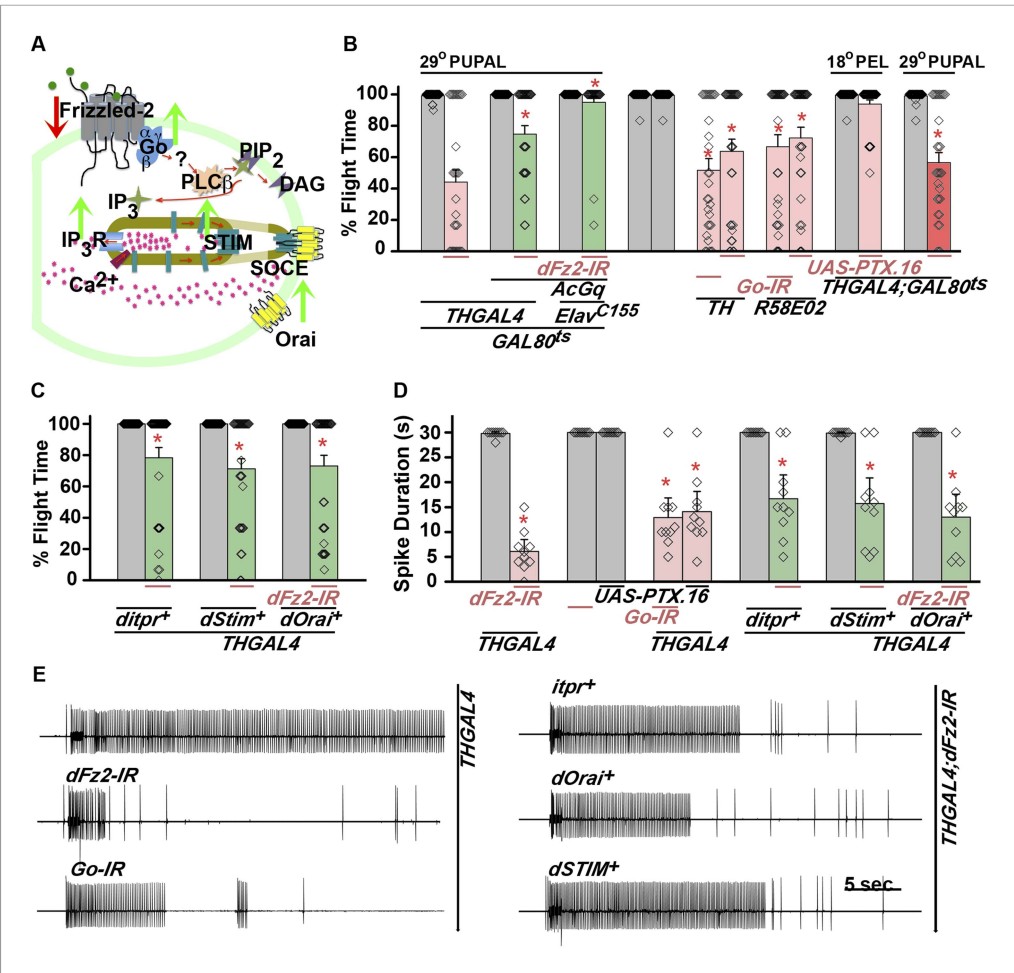

**Figure 3**. dFz2 function is mediated through G-protein Go and IP₃-mediated calcium signaling in dopaminergic neurons. (**A**) A schematic showing dFz2-mediated activation of Go followed by IP₃R-mediated $Ca^{2+}$ signaling pathway and Store-operated $Ca^{2+}$ entry (SOCE) through dSTIM and dOrai. Red (down-regulation) and green (over-expression) arrows indicate the two strategies used for testing this signaling mechanism. (**B**) Percentage flight times of the indicated genotypes are shown. Knockdowns flight times were compared to their respective heterozygote controls, whereas AcGo rescue of dFz2 knockdown was compared to dFz2 knockdown (*p < 0.001, Mann–Whitney U-test). (**C**) Percentage flight times of heterozygous controls (gray bars) followed by over-expression of calcium signaling molecules (itpr+, dStim+, dOrai+) in flies with knockdown of dFz2 (green bars). Overexpression of calcium signaling molecules (itpr+, dStim+, dOrai+) rescued flight defects significantly when compared to flies with dFz2 knockdown (*p < 0.001, Mann–Whitney U-test). (**D**) Durations of rhythmic action potentials recorded from the DLMs of air-puff stimulated tethered flies. Spike durations were reduced upon expression of Go RNAi or UAS-PTX.16 in dopaminergic neurons and partially rescued upon over-expression of calcium signaling molecules (itpr+, dStim+, dOrai+) when compared to knockdown of dFz2 (*p < 0.001, Mann–Whitney U-test). (**E**) Representative electrophysiological recordings from DLMs of the indicated genotypes.

The following figure supplements are available for figure 3:

**Figure supplement 1**. The canonical Fz2/β catenin signaling pathway does not function downstream of dFz2 in the context of flight circuit maturation.

**Figure supplement 2**. Go functions downstream of dFz2 in the context of flight circuit maturation.

**Figure supplement 3**. Non-canonical dFz2/$Ca^{2+}$ signaling functions downstream of dFz2 in the context of flight circuit maturation.

through the $IP_3R$ leads to clustering of dSTIM, which in turn promotes Store-operated $Ca^{2+}$ entry (SOCE) through dOrai (*Venkiteswaran and Hasan, 2009*; *Agrawal et al., 2010*) (*Figure 3A*). In a converse experiment, we tested the effect of over-expression of a $dFz2^+$ transgene on flight deficits induced by pan-neuronal knockdown of the $IP_3R$ (*itpr-IR)*, dSTIM (*dSTIM-IR*) and dOrai (*dOrai-IR*) and observed a significant rescue in all three conditions tested (*Figure 3—figure supplement 3*, *Video 3*, and *Video 4*). These data suggest that, as in vertebrate neurons, dFz2 links to intracellular calcium signaling in *Drosophila*. Next, we tested the effect of over-expression of $dSTIM^+$, on flight deficits induced by *dFz2* knockdown in dopaminergic neurons. A partial but significant rescue of flight was observed (*Figure 3C*) accompanied by a rescue of the duration of firing patterns from the DLMs (*Figure 3D,E*). Over-expression of the $IP_3R$ and dOrai also rescued flight deficits observed by knockdown of dFz2 in dopaminergic neurons (*Figure 3C,D,E*). Together these data support the idea that maturation of dopaminergic neurons in the flight circuit requires intracellular $Ca^{2+}$ signaling by activation of dFz2 and Go. The mechanism by which Go activates $Ca^{2+}$ signaling through $IP_3R$/dSTIM requires further investigation (*Figure 3A*).

## Down-regulation of dFz2 affects neuronal activity in maturing PAM neurons required for adult flight

Synaptic function of developing hippocampal neurons can be modulated by $Ca^{2+}$ signaling downstream of Fz2 (*Varela-Nallar et al., 2010*). In *Drosophila*, neuronal activity can be increased by over-expression of a voltage-gated sodium channel, *NaChBac* (*Nitabach et al., 2006*). Therefore, we tested flight in organisms with *dFz2* knockdown and increased neuronal activity by *NaChBac* expression. Flight was restored close to 100% upon expression of *NaChBac* in dopaminergic neurons (*THGAL4*) and more specifically in PAM neurons (*NP6510GAL4 and R58E02GAL4*; *Figure 4A*, *Video 5*). Moreover, raising neuronal activity during pupal development in parallel with *dFz2* knockdown compensated for loss of flight observed in the knockdown condition (*Figure 4B*). These data suggest that $Fz2/Ca^{2+}$ signaling can contribute to the synaptic activity of dopaminergic PAM neurons in pupae. The requirement for synaptic activity in maturing PAM neurons was tested directly by expression of a temperature sensitive mutant of the dynamin orthologue, *shibire^{ts}* (*Shi^{ts}*). Expression of *Shi^{ts}* blocks vesicle endocytosis at 30°C (*Kitamoto, 2001*) and its expression during pupal development, either in TH neurons (*THGAL4*) or exclusively in PAM neurons (*NP6510GAL4 and R58E02GAL4*) resulted in significant loss of flight (*Figure 4C*). Temporal expression of *Shi^{ts}* in adult PAM neurons also resulted in a flight deficit (*Figure 4D*), supporting the requirement of active synaptic transmission in PAM neurons for adult flight.

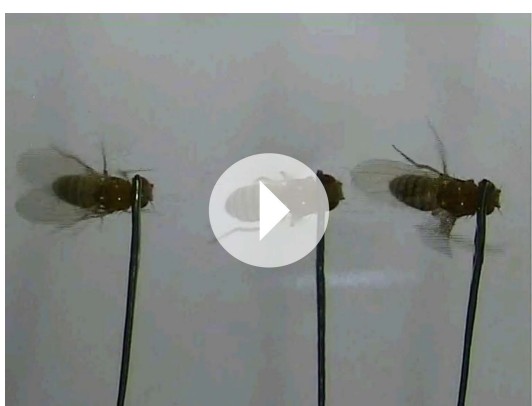

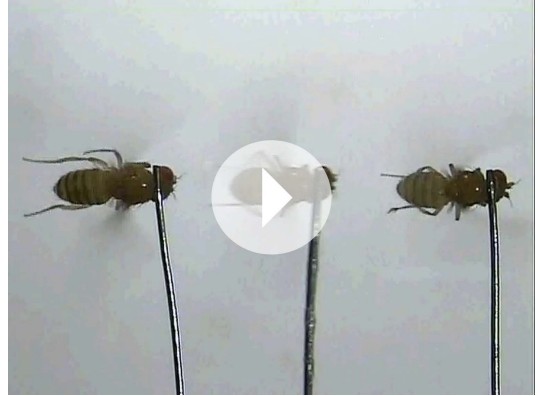

**Video 3.** Overexpression of $IP_3R$ in dopaminergic neurons rescues flight defects of dFz2 downregulation. Real time video recording of air-puff induced flight in the following genotypes from left to right. (1) *THGAL4; dFz2-IR;itpr^+*, (2) *THGAL4;dFz2-IR*, (3) *dFz2-IR/+*. Following a gentle air-puff *THGAL4; dFz2-IR; itpr^+* flies were able to initiate and maintain flight for a longer duration as compared to *THGAL4;dFz2-IR*.

**Video 4.** Flight defects in dFz2 knockdown individuals can be rescued by over-expression of dSTIM. Real time video recording of air-puff induced flight in the following genotypes from left to right. 1) *THGAL4; dFz2-IR; dSTIM^+*, 2) *THGAL4; dFz2-IR*, 3) *dFz2-IR/+*. Following a gentle air-puff *THGAL4; dFz2-IR; dSTIM^+* flies were able to initiate and maintain flight for a longer duration as compared to *THGAL4; dFz2-IR*.

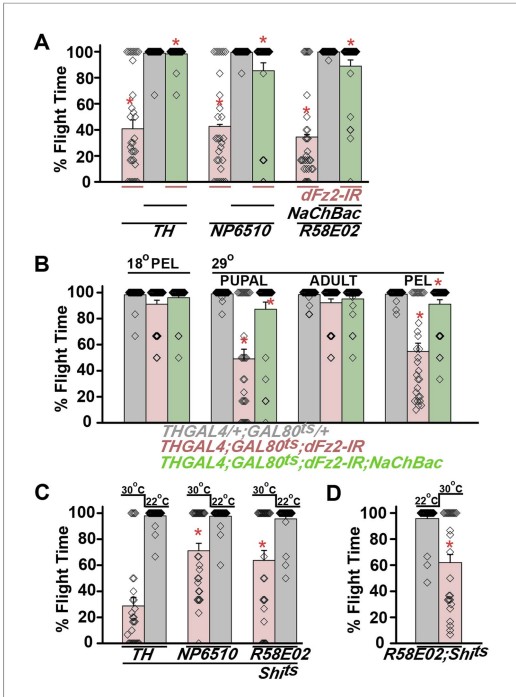

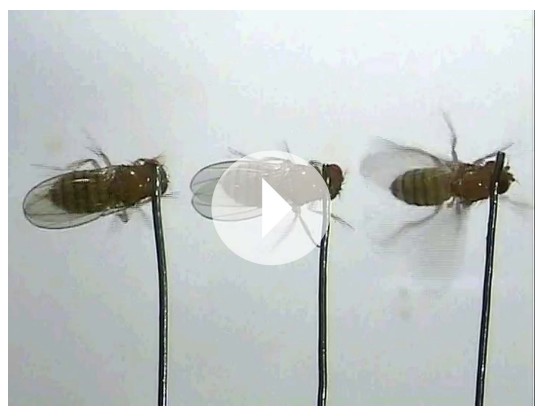

**Video 5.** Increased neuronal activity in PAM neurons rescues flight in individuals with dFz2 knockdown. Real time video recording of air-puff induced flight in the following genotypes from left to right. (1) *R58E02GAL4; dFz2-IR;NaChBac*, (2) *R58E02GAL4;dFz2-IR*, (3) *dFz2-IR/+*. Following a gentle air-puff *R58E02GAL4; dFz2-IR; NaChBac* flies were able to initiate and maintain flight for a longer duration as compared to *R58E02GAL4; dFz2-IR*.

**Figure 4**. Knockdown of dFz2 affects neuronal activity of maturing flight circuit PAM neurons. (**A**) Percentage flight times of individual heterozygous controls (gray bars), *dFz2* knockdown (*dFz2-IR*) in dopaminergic neurons (*THGAL4*) and PAM neurons (*NP6510GAL4, R58E02GAL4*) (red bars) followed by over-expression of *NaChBac* in presence of *dFz2-IR* (green bars); (*p < 0.001, Mann–Whitney U-test). (**B**) Percentage flight times for heterozygotes of *THGAL4;GAL80^ts* (gray bars) followed by stage-specific knockdown of *dFz2* (red bars) and over-expression of *NaChBac* in flies with *dFz2* knockdown (green bars) as indicated. Over-expression of *NaChBac* during pupal development rescued flight as did over-expression post-egg laying (PEL) (*p < 0.001, Mann–Whitney U-test). (**C**) Percentage flight times upon expression of *Shibire^ts* (*Shi^ts* 30°C; red bars) either in pupal or no expression (*Shi^ts* 22°C; gray bars). Expression was either in dopaminergic neurons (*THGAL4*) or PAM neurons (*NP6510GAL4, R58E02GAL4*). Flight was tested at 25°C. Expression of *Shi^ts* in pupal resulted in reduced flight times. (**D**) Percentage flight times upon adult expression of *Shibire^ts* (*Shi^ts* 30°C; red bars) or no expression (*Shi^ts* 22°C; gray bars), in PAM neurons with *R58E02GAL4*. Flight was tested at 30°C. Expression of *Shi^ts* resulted in reduced flight times (*p < 0.001, Mann–Whitney U-test).

## dFz2 is required in PAM neurons for normal expression of TH

The cellular effect of reduced *dFz2* expression in PAM neurons was investigated next. TH levels in PAM neurons marked by *R58E02GAL4*, appear reduced upon *dFz2* knockdown as judged by immunohistochemistry (compare anti-TH panels in *Figure 5A,B*). Expression of *NaChBac* with *dFz2-IR* restored TH expression close to wild-type levels (*Figure 5A,B,C*). Quantification of TH immunostaining across multiple samples revealed a significant reduction upon *dFz2* knockdown which was restored by expression of *NaChBac* (*Figure 5D,E*). Furthermore, *TH* transcript levels were significantly reduced by *dFz2* knockdown and were restored upon expression of *NaChBac* (*Figure 5—figure supplement 1*). Thus, altered TH levels corroborated well with flight deficits and their rescue in various genotypes. Numbers of TH-positive neurons in the PAM cluster were not significantly different between the three genotypes as judged by anti-TH immunostaining (*Figure 5G*). Surprisingly, *R58E02GAL4*-driven GFP expression was also reduced upon expression of *dFz2-IR* and was restored back upon expression of *NaChBac* (*Figure 5—figure supplement 1*). Consequently, there was an apparent reduction in the numbers of GFP positive cells upon *dFz2-IR* expression which was restored by *NaChBac* expression (*Figure 5F*). Because the number of *TH*-expressing cells of the PAM cluster remained unchanged upon *dFz2* knockdown and after *NaChBac* rescue (*Figure 5G*), we hypothesized that dFz2/Ca$^{2+}$ signaling regulates *TH* expression in PAM neurons during pupal development.

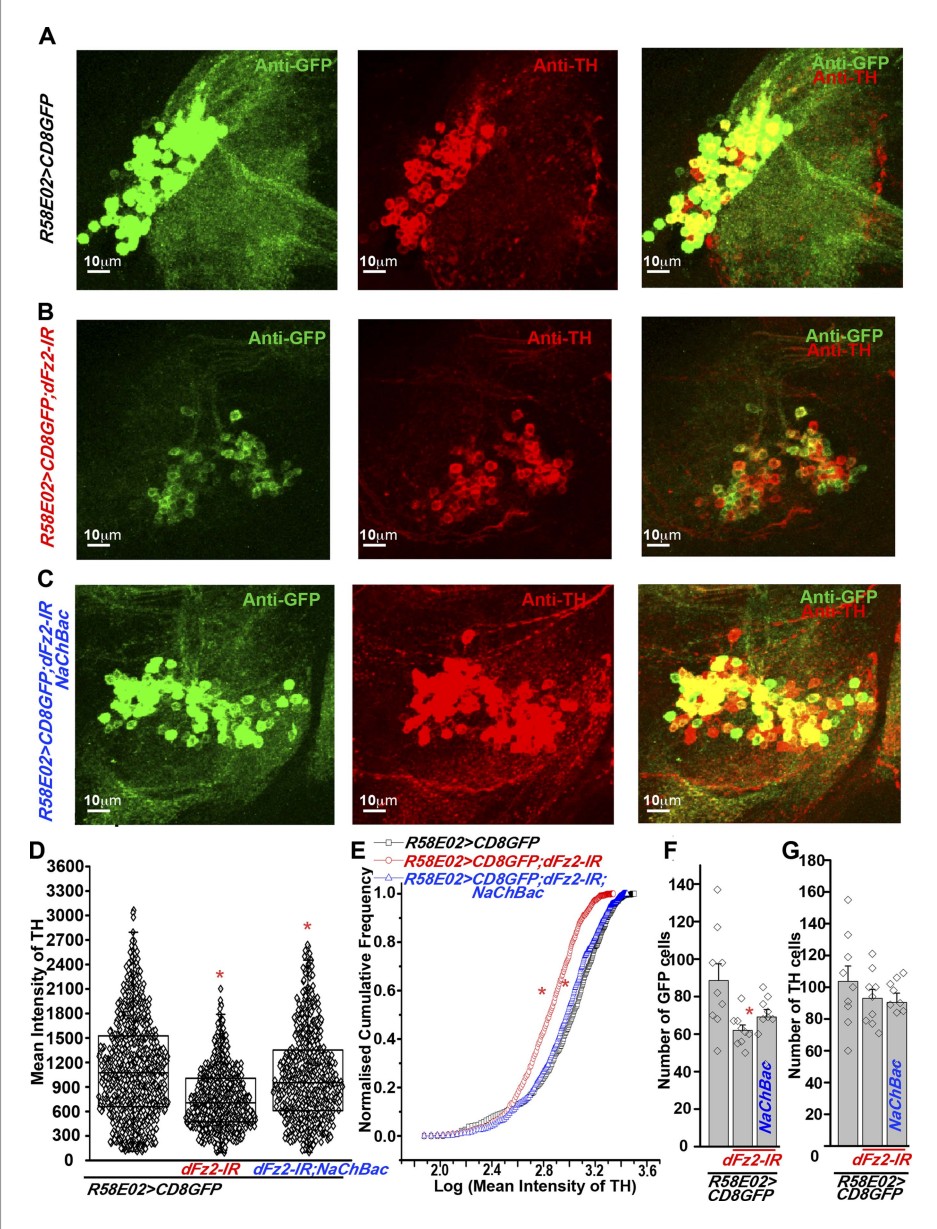

**Figure 5**. Expression of TH is reduced in PAM neurons by dFz2 knockdown. (**A**) Expression of GFP (Anti GFP; green) and TH (Anti TH; red) is shown in PAM dopaminergic neurons marked by *R58E02GAL4>mCD8GFP*. (**B**) Significant reduction of GFP and TH immunoreactivity is observed in PAM neurons of *R58E02GAL4>mCD8GFP; dFz2-IR* individuals; which is (**C**) rescued by over-expression of *NaChbac* (*R58E02GAL4> mCD8GFP; dFz2-IR;NaChBac*). (**D**) Scatter plot with the mean intensity of TH expression in individual PAM neurons (N = 1280) in the indicated genotypes. Cells were obtained from 16 brain hemispheres; *p < 0.05, one-way ANOVA. (**E**) A Kolmogorov-Smirnov (K-S) plot analyzing the distribution of the mean intensity of TH immunoreactivity in PAM neurons. The frequency distribution is significantly shifted to the left for *R58E02GAL4>mCD8GFP;dFz2-IR* as compared to *R58E02GAL4>mCD8GFP* indicating a significantly higher percentage of cells with lower mean intensity. Frequency distribution of *R58E02GAL4> mCD8GFP; dFz2-IR; NaChBac* is shifted back towards the control distribution *R58E02GAL4>mCD8GFP*, indicating a significant rescue (*$p_{K-S}$ < 0.05). (**F**) Total number of GFP positive cells and (**G**) TH positive cells were counted in the indicated genotypes. No difference in the number of TH cells was found; however GFP cells were reduced upon *dFz2* knockdown (*p < 0.05, one-way ANOVA).

The following figure supplements are available for figure 5:

**Figure supplement 1**. Expression of GFP is altered upon expression of *dFz2-IR* in PAM neurons using *R58E02GAL4*.

*Figure 5. continued on next page*

*Figure 5. Continued*

**Figure supplement 2**. Altered GFP expression was seen upon expression of *dFz2-IR* in dopaminergic neurons.

**Figure supplement 3**. Knockdown of dFz2 in *OK371GAL4*-expressing neurons does not affect TH expression-positive PAM neurons.

Moreover, our data support a role for dFz2/Ca$^{2+}$ signaling in regulating expression of the *R58E02GAL4* transgene where GAL4 is under control of the *fumin* gene encoding a Dopamine Transporter, DAT (*Liu et al., 2012*).

The status of TH and GFP expression in PAM neurons was further investigated after *dFz2* knockdown with *THGAL4*. Unlike *R58E02GAL4*, *THGAL4*-driven mGFP marks a small subset of PAM neurons. This consists of two clusters of 6–7 neurons each (**Figure 2A** and **Figure 5—figure supplement 2**) (*Riemensperger et al., 2013*). *THGAL4*-driven expression of *dFz2-IR* resulted in significant loss of GFP expression in PAM neurons (**Figure 5—figure supplement 2**). Interestingly, GFP expression in five TH-positive neurons of the PAL cluster remained unaffected by knockdown of dFz2, suggesting that dFz2 regulation of TH expression maybe PAM specific (data not shown). Over-expression of either *dSTIM+*, *IP$_3$R* or *NaChBac* in the background of *dFz2* knockdown could partially rescue loss of GFP expression in the PAM neurons (**Figure 5—figure supplement 2**). Thus, *THGAL4*-driven GFP expression in PAM neurons correlated with flight deficits and their rescue (**Figures 1A, 3C, 4A**). Quantification of *TH* immunoreactivity in PAM cells by *THGAL4*-driven dFz2 knockdown was technically not possible because the few *THGAL4*-positive cells of the PAM cluster could not be identified in the *dFz2* knockdown condition. Taken together these data support the idea that dFz2/Ca$^{2+}$ signaling in PAM neurons drives transcription of two key dopamine synthesis and uptake molecules, TH and DAT. The transcriptional regulation extends to GAL4 transgenic constructs containing TH and DAT regulatory sequences. As controls we tested TH immunoreactivity of PAM neurons in flies with dFz2 knockdown in glutamatergic neurons (*OK371GAL4*). Both TH immunoreactivity of PAM neurons (**Figure 5—figure supplement 3**) and flight patterns (**Figure 1—figure supplement 1**) were similar to controls.

## Altered levels of TH in the dopaminergic PAM neurons cause flight deficits

Based on our observation that *dFz2* knockdown in PAM neurons leads to flight deficits accompanied by a significant reduction of TH expression, we tested the requirement of TH in PAM neurons for flight. Over-expression of a neuronal-specific TH cDNA transgene (*UASDTH1*) (*Friggi-Grelin et al., 2003*) in flies with *dFz2* knockdown by *PAMGAL4* strains (*NP6510GAL4 and R58E02GAL4*) could rescue flight deficits significantly (**Figure 6A**, **Video 6**, **Figure 6—figure supplement 1**). Furthermore, knockdown of TH with an RNAi (*dTH-IR*) resulted in significant loss of flight and reduced TH expression (**Figure 6A–C,F**, **Figure 6—figure supplement 1**). Moreover, knockdown of TH in PAM neurons affected *R58E02GAL4*-driven GFP expression suggesting feedback regulation of dopamine transporter (DAT) by dopamine levels. Over-expression of the *DTH1* neuronal cDNA could rescue TH immunoreactivity in the *R58E02GAL4*-expressing PAM neurons with *dFz2* knockdown (**Figure 6B,C, E**). However, GFP immunoreactivity remained low and unchanged between *dFz2-IR*- and *dFz2-IR*; *DTH1*-expressing PAM neurons (**Figure 6—figure supplement 1**). These data suggest that rescue of flight by over-expression of *DTH1* by passes the transcriptional regulation of *DAT* by dopamine and of endogenous *TH* by dFz2/Ca$^{2+}$ signaling. They confirm the requirement for TH expression in PAM neurons for flight.

## Maintenance of acute flight requires synaptic activity in α′ β′ lobes of the mushroom body

PAM neurons send projections to the horizontal lobes of the MB neuropil (*Aso et al., 2012*; *Burke et al., 2012*; *Liu et al., 2012*; *Riemensperger et al., 2013*). The MB is a paired brain structure that controls several higher brain functions in insects ranging from olfactory memory formation and reinforcement (*Kahsai and Zars, 2011*; *Waddell, 2013*) to locomotor activity (*Helfrich-Forster et al., 2002*; *Serway et al., 2009*; *Riemensperger et al., 2013*). Axons and dendrites of Kenyon cells, positioned in the calyx region form the MB neuropil which is subdivided into the α, β, α′, β′, and γ lobes

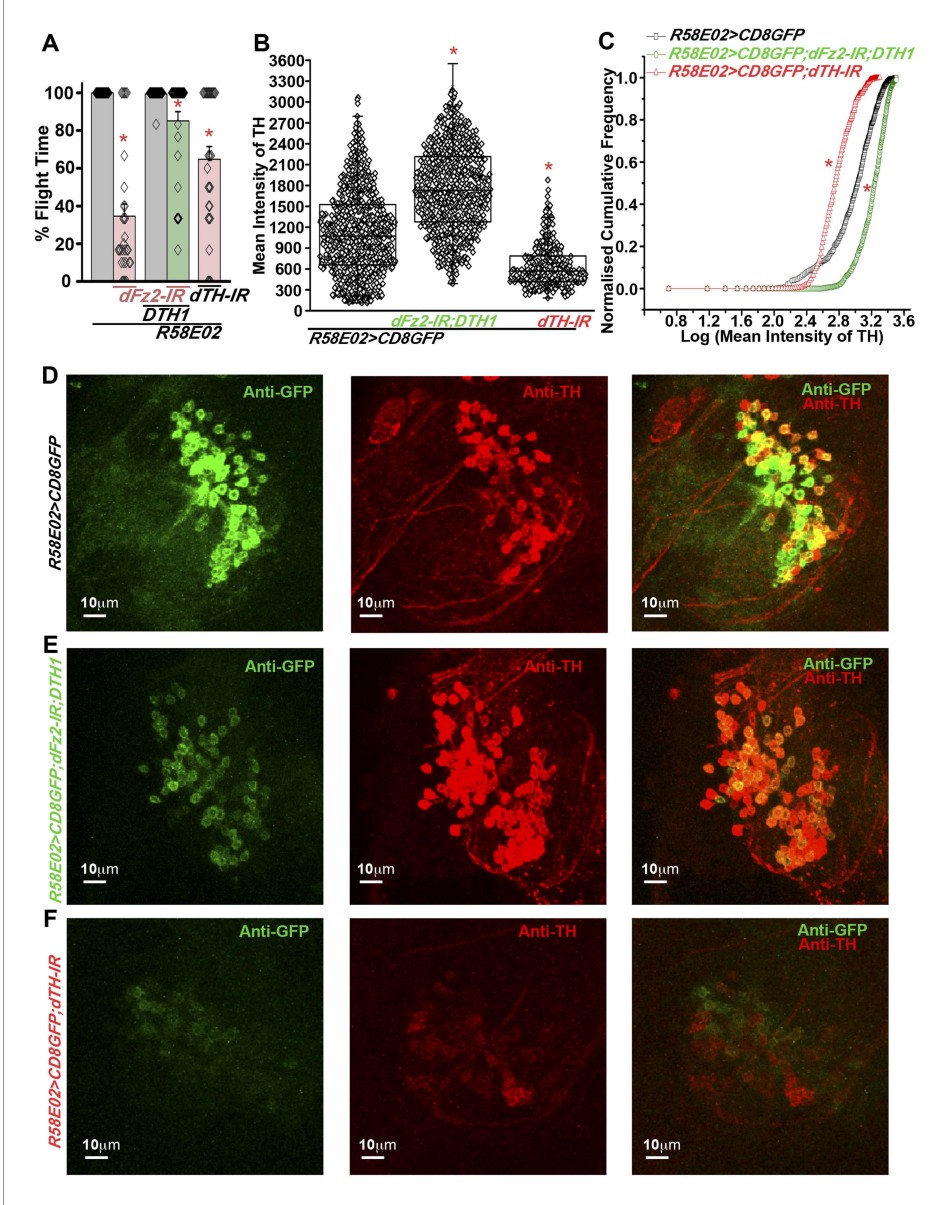

**Figure 6**. Expression of *DTH1* in PAM neurons rescues flight defects shown by dFz2 knockdown. (**A**) Percentage flight times of individual heterozygous control flies (gray bars), flies with expression of *dFz2-IR* and *THRNAi* (*dTH-IR*) in PAM neurons (*R58E02GAL4*) (red bars), and flies with over-expression of *DTH1* in the presence of *dFz2-IR* (green bars). Expression of *DTH1* rescued the flight defect of dFz2 knockdown flies to a significant extent (*p < 0.001, Mann–Whitney U-test). (**B**) Scatter plot of the mean intensity of TH expression in individual PAM neurons (N = 1280) from 16 brain hemispheres of the indicated genotypes (*p < 0.05, one-way ANOVA). (**C**) Kolmogorov–Smirnov (K-S) plot analyzing the distribution of the cellular mean intensity shown in **B**. The frequency distribution is significantly shifted to the left for *R58E02GAL4> mCD8GFP;dTH-IR* as compared to *R58E02GAL4>mCD8GFP* indicating a higher number of cells with lower mean intensity of TH. Frequency distribution of *R58E02GAL4> mCD8GFP; dFz2-IR;DTH1* is shifted back to the right indicating fewer cells with lower mean intensity (*$p_{K-S}$ < 0.05). (**D**) Expression of GFP (Anti GFP; green) and TH (Anti TH; red) is shown in PAM dopaminergic neurons in *R58E02GAL4>mCD8GFP*, (**E**) *R58E02GAL4>mCD8GFP; dFz2-IR; DTH1* and (**F**) *R58E02GAL4> mCD8GFP;dTH-IR*.

The following figure supplement is available for figure 6:

**Figure supplement 1**. Expression of GFP is altered in PAM neurons upon knockdown of *dFz2* in the presence of either *DTH1* or *dsDTH*.

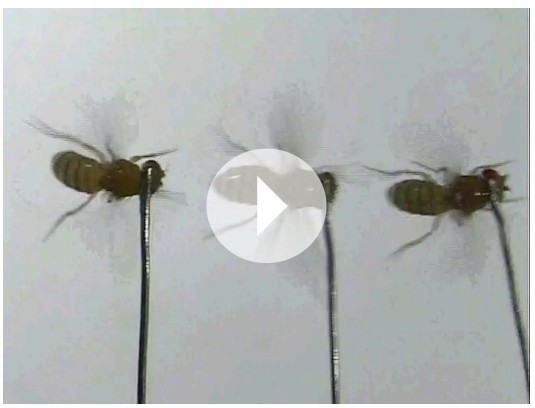

**Video 6.** DTH over-expression in PAM neurons rescues flight in individuals with dFz2 knockdown. Real time video recording of air-puff induced flight in the following genotypes from left to right. (1) *R58E02GAL4; dFz2-IR;DTH1*, (2) *R58E02GAL4;dFz2-IR*, (3) *dFz2-IR/+*. Following a gentle air-puff *R58E02GAL4; dFz2-IR;DTH1*, flies were able to initiate and maintain flight for a longer duration as compared to *R58E02GAL4;dFz2-IR*.

(*Strausfeld et al., 2003*). We reasoned that flight deficits observed due to reduced levels of TH in PAM neurons might derive from reduced dopamine release and signaling in postsynaptic MB neurons. This idea was tested by silencing specific MB neuropil lobes with *mb186bGAL4* and *mb247GAL4* drivers. *mb186bGAL4* is a recently generated split GAL4 strain (*Aso et al., 2014*) whose expression is restricted to the α′ β′ lobes (*Vogt et al., 2014*), whereas *mb247GAL4* is expressed in the α, β, and γ lobes (*Zars, 2000*; *Krashes et al., 2007*; *Pech et al., 2013*). Synaptic release in MB neurons of adult flies was inhibited by expression of a temperature sensitive dominant negative dynamin transgene (*UAS-Shi^{ts}*) under control of either *mb186bGAL4* or *mb247GAL4* drivers. Blocking synaptic release in α′, β′ lobes resulted in a strong flight deficit, whereas silencing of the α, β , and γ lobes did not have a significant effect on flight, for 30 s (*Figure 7A*). Reduced flight bouts were accompanied by loss of rhythmic spiking in the DLMs of flies with silenced α′, β′ lobes neurons (*Figure 7B*). These data support a requirement for post-synaptic dopamine receptors in MB neurons that function for maintenance of acute flight. We tested this requirement further by RNAi-mediated knockdown of the four dopamine receptors—*DopECR (CG18314)*, *Dop1R1 (CG9652)*, *Dop1R2 (CG18741)*, and *Dop2R (CG33517)*, in either the α′, β′ neurons (*mb186bGAL4*) or the α, β, γ neurons (*mb247GAL4*; *Figure 7—figure supplement 1*). A reduction in the length of flight bouts was observed specifically upon knockdown of Dop1R2 in the α′, β ′ neurons (*Figure 7A*). The role of α′ β′ lobes, in flight was supported by another GAL4 driver, *c305GAL4*, which expresses in the α′ β′ lobes and faintly in the γ lobe (*Krashes et al., 2007*; *Pech et al., 2013*). Blocking synaptic activity or knockdown of Dop1R2 using *c305aGAL4* resulted in significant flight deficits (*Figure 7—figure supplement 2*).

Unlike dopaminergic neurons located in the ventral ganglion, which directly modulate flight motor neuron function as demonstrated recently (*Sadaf et al., 2015*), the PAM–MB circuit described here is not known to project to flight motor neurons in the ventral ganglion (*Riemensperger et al., 2013*). Rather, PAM-MB circuits function to reinforce both aversive and appetitive olfactory responses (*Waddell, 2013*). To test possible re-inforcement of flight time by the PAM-MB circuit, we monitored longer flight bouts in several genotypes. Knockdown of *dFz2* in PAM neurons (*R58E02GAL4>dFz2-IR*) significantly reduced the duration of flight bouts, monitored up to 15 min, from an average of 13.04 ± 0.4 min in controls to 0.85 ± 0.3 min in the knockdowns (*Figure 7C*). Maintenance of flight bouts was rescued significantly upon increasing neuronal activity by expression of *NaChBac* (*Figure 7C*). These data were analyzed further by binning flight bouts in 20-s intervals (*Figure 7D*). Flight time of *dFz2* knockdown flies clustered towards the left among shorter flight bouts, whereas control flies clustered towards the right with longer flight bouts (*Figure 7D*). Distribution of flight times in *NaChBac*-rescued flies appeared intermediate. All rescued flies flew for longer than 20 s and a small percentage flew for longer than 10 min. Interestingly, *DTH1*-rescued flies exhibit shorter flight bouts as compared with *NaChBac*-rescued organisms (*Figure 7C,D*). This difference in rescue abilities may in part be due to the previous observation that *NaChBac* rescue restores TH immunoreactivity to PAM neurons (*Figure 5C*), whereas TH rescue very likely cannot restore the excitability deficit of PAM neurons. Long flight bouts were also tested in flies with *dFz2* knockdown by *THGAL4*, followed by rescue with *dSTIM^+*, *itpr^+*, and *NaChBac* (*Figure 7—figure supplement 4*). The *NaChBac* rescue profile was very similar to that observed by *NaChBac* rescue of PAM-specific *dFz2* knockdown, whereas, *dSTIM^+* and *itpr^+* rescue profiles resembled the *DTH1* rescue in *Figure 7C,D* (*Figure 7—figure supplement 5*). These data support a role for the PAM-MB circuit in maintenance of long flight bouts through dopaminergic synapses on α′ β′ MB lobes.

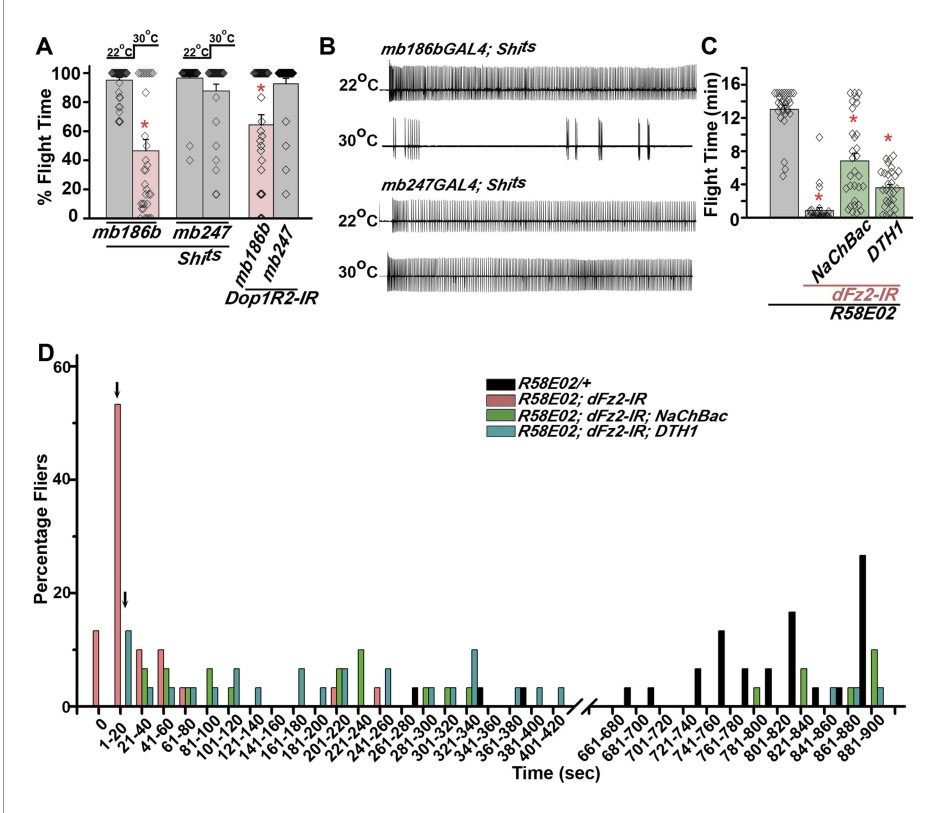

**Figure 7**. Mushroom body α′/β′ neurons regulate flight through Dop1R2. (**A**) Percentage flight times of individual flies of the indicated genotypes. Flight defects were seen by reducing the activity of α′/β′ neurons (*mb186bGAL4*, red bar) and by knockdown of *Dop1R2* in mushroom body α′/β′ neurons (*p < 0. 01, Mann–Whitney U-test). (**B**) Electrophysiological responses from the DLMs showed similar responses as observed during flight. (**C**) Flight times during longer flight tests monitored over 15 min are shown. Over-expression of *NaChBac* rescued flight time partially when compared to knockdown to *dFz2* (*p < 0.001, Mann–Whitney U-test). (**D**) Percentage of flies that either do not initiate flight (0 s) or fly for time-periods within the binned intervals (20 s each) is shown for the indicated genotypes.

The following figure supplements are available for figure 7:

**Figure supplement 1**. Dopamine receptor knockdown in MB neurons.

**Figure supplement 2**. Synaptic activity in α′/β′ lobes required for flight.

**Figure supplement 3**. Knockdown of dFz2 does not affect climbing ability of flies.

**Figure supplement 4**. Maintenance of flight requires dFz2/Ca²⁺ signaling in dopaminergic neurons.

**Figure supplement 5**. Maintenance of flight requires Fz2/Ca²⁺ signaling in dopaminergic neurons.

## Discussion

### dFz2 signaling maintains TH levels in the dopaminergic PAM cluster

Differentiation of neuronal subtypes, after genetic specification, is subject to multiple signals many of which generate and modify electrical activity of the cognate neurons (*Borodinsky et al., 2014*). We demonstrate a requirement for dFz2/Ca²⁺ signaling for maintaining TH levels in a subset of central brain dopaminergic neurons—the PAM cluster. Our results support transcriptional regulation of *TH* and very likely the dopamine transporter (DAT) by dFz2/Ca²⁺ signaling in the

PAM neurons. A significant compensation of the flight deficit was observed in flies with *dFz2* knockdown in PAM neurons upon over-expression of the sodium channel *NaChBac*, indicating that dFz2/Ca$^{2+}$ signaling also affects neural activity of PAM neurons. Moreover, flight deficits were observed upon expression of *Shi$^{ts}$* in PAM neurons during the pupal stages, supporting a role for neural activity and synaptic transmission in their development. Increased *TH* transcripts and TH immunoreactivity after rescue by *NaChBac* suggests that dFz2/Ca$^{2+}$ signaling can in part be compensated by raised neural activity, and possibly the two signaling mechanisms function in parallel for maintaining *TH* transcription in PAM neurons. It is likely that in addition to TH and DAT, dFz2/Ca$^{2+}$ signals exert their influence on other transcripts in PAM neurons. Transcriptional profiling of the PAM neurons is necessary to address this possibility. Interestingly, despite an increase in the number of PAM neurons during pupal maturation (*Figure 2—figure supplement 3*), we do not observe an affect of dFz2/Ca$^{2+}$ signaling on the number of PAM neurons. Knockdown of dFz2 by *DdcGAL4* that marks >60 PAM neurons resulted in flight time of ~65% (*Figure 1A*), whereas a stronger flight deficit was observed with *NP6510GAL4* which marks just 15 PAM neurons (*Figure 2B*). Moreover, the flight deficit obtained with *R58E02GAL4* which marks ~100 PAM neurons was similar to the flight deficit obtained with *NP6510GAL4* (*Figure 2B*). Thus, the numbers of PAM neurons that express *dFz2-IR* do not correlate with the extent of observed flight deficits, suggesting that flight is regulated by a subset of PAM neurons and their projections to the MB. Further, analysis with GAL4 strains that mark PAM neuronal sub-domains would be helpful in identifying such flight specific PAM neurons.

Down-regulation of GFP fluorescence in *PAMGAL4* strains upon *dFz2* knockdown prevented direct analysis of their projections to the MB. Based on a recent study demonstrating similar PAM-MB connections for negative geotaxis (*Riemensperger et al., 2013*), we measured climbing in *PAM > dFz2-IR* flies. This appeared similar to controls (*Figure 7—figure supplement 3*), supporting the idea that connections of PAM neurons to the MB are maintained upon dFz2 knockdown. Thus as compared to climbing, flight appears more sensitive to the observed imbalance of *TH*. However, as expected inhibition of synaptic release from PAM neurons (*R58E02GAL4>Shi$^{ts}$*; at 30°C) affected both flight (*Figure 4D*) and climbing (*Figure 7—figure supplement 3*).

## dFz2 and calcium signaling in *Drosophila*

In vertebrates, βγ subunits of the trimeric G$_o$ protein activate phospholipase Cβ, which in turn enhances IP$_3$ formation followed by IP$_3$ receptor-mediated release of calcium from endoplasmic reticular stores (*Rebecchi and Pentyala, 2000*). In *Xenopus* embryos, non-canonical Wnt/Ca$^{2+}$ signaling, acting through Fz receptors, activates the Nuclear Factor of Activated T cells (NFAT) which regulates transcription of genes required for dorsoventral axis formation (*Saneyoshi et al., 2002*). Apart from NFAT, non-canonical Fz2/Ca$^{2+}$ signaling can also activate calcium calmodulin-dependent protein kinase II (CamKII) and protein kinase C (PKC) which regulate activity of transcription factors, such as NFκB and CREB (*Sheldahl et al., 1999*; *Kuhl et al., 2000*; *Slusarski and Pelegri, 2007*). Non-canonical dFz2/Ca$^{2+}$ signaling has been poorly characterized in *Drosophila*. dFz2 can be cleaved and imported into the nucleus in *Drosophila* neurons (*Mosca and Schwarz, 2010*). However, we do not favor direct transcriptional control of *TH* by cleaved dFz2 for the following reasons. Rescue of flight in *dFz2* knockdown flies can be achieved by *AcGo*, *itpr$^+$*, and *dSTIM$^+$* (*Figure 2*). These data support a link between dFz2 activation of Go at the membrane followed by intracellular Ca$^{2+}$ release through the IP$_3$R and dSTIM-mediated calcium entry. This mechanism is broadly similar to what has been observed in vertebrates. Moreover, we did not detect nuclear dFz2 in PAM neurons (*Figure 2—figure supplement 2*).

Our observation that knockdown of *dFz2* reduced not only transcripts from the endogenous TH gene but also affected GFP expression from a DAT promoter transgene (*R58E02GAL4>GFP*) suggests co-ordinated transcriptional regulation of genes for maintenance of dopamine levels by Fz2/Ca$^{2+}$ signaling in PAM neurons. However, the molecular mechanism by which reduced Fz2/Ca$^{2+}$ signaling regulates transcription of *TH* and very likely other genes, in PAM neurons remains to be elucidated.

## PAM-MB connectivity and flight

Insect MBs are lobed structures located bilaterally in the protocerebrum of the central nervous system. Neuro-anatomical studies have demonstrated the presence of both efferent neurons arising

from MB lobes as well as afferent connections supplying the MB lobes from protocerebral regions (*Ito et al., 1998*) including dopaminergic innervations from the PAM neurons (*Mao and Davis, 2009*). The *Drosophila* MB has been studied extensively as a central brain hub for olfactory associative memory and behavior (*Heisenberg, 2003*; *Fiala, 2007*). PAM-MB circuits are required for aversive as well as rewarding reinforcement of olfactory information (*Aso et al., 2012*; *Burke et al., 2012*; *Liu et al., 2012*). Further analysis of our data revealed that majority (~55%) of flies with *dFz2* knockdown in the PAM neurons fly for 1–20 s as compared to controls that can fly for 700–900 s or more. Rescue of flight deficits in flies with *dFz2* knockdown in PAM neurons either by over-expression of a sodium channel (*NaChBac*) or a transgene encoding TH (*UAS-DTH1*) supports a requirement for both neural activity and dopamine release from the PAM neurons for maintenance of longer flight bouts (*Figure 7*). Reduced flight times are very likely due to lack of dopaminergic reinforcement during flight arising from reduced strength of PAM-MB signaling.

The role of higher brain centres in *Drosophila* flight has been investigated primarily in the context of visual cues, and these studies identified the central complex as a key area for visual associative learning (*Ofstad et al., 2011*). The flight circuit identified here appears similar to the one identified recently for the startle induced climbing response which requires PAM dopaminergic inputs to the β' lobe (*Riemensperger et al., 2013*). Taken together, our findings support an emerging role for the *Drosophila* MB in coordinated motor behavior, previously considered unlikely (*Wolf et al., 1998*). Dopaminergic inputs from the PAM to the MBs might help integrate olfactory sensory information with motor behavior essential in a natural environment. Further investigations should allow a better understanding of how MB centres for re-inforcement of olfactory memory interact with the flight motor system.

## Materials and methods

### Fly rearing and stocks

*Drosophila* was reared on corn flour/agar media supplemented with yeast, grown at 25°C, unless otherwise mentioned in the experimental design. The pan-neuronal GAL4 driver (*Elav$^{C155}$GAL4*), aminergic GAL4 (*DdcGAL4*) (*Li et al., 2000*) and mushroom body drivers *c305aGAL4* and *mb247GAL4* were obtained from Bloomington Stock Center, Bloomington, IN. *mb186bGAL4* was obtained from Anja Beatrice Freidrich (MPG, Germany). The dopaminergic GAL4 (*THGAL4*), serotonergic GAL4 (*TRHGAL4*), and two other GAL4s, *NP6510GAL4* and *R58E02GAL4* were generously provided by Serge Birman (CNRS, ESPCI Paris Tech, France) (*Riemensperger et al., 2013*). The various dopaminergic subdomain GAL4 drivers used, *THC'GAL4*, *THC1GAL4*, and *THF2GAL4*, were obtained from Mark N Wu (Johns Hopkins University, Baltimore) (*Liu et al., 2012*). The peptidergic GAL4 (*P386GAL4*) was obtained from Paul Taghert (Washington University, St. Louis) (*Taghert et al., 2001*).

UAS strains of Frizzled-2 RNAi (*9739R-1(II)*, referred to as *dFz2-IR* in the text and figures) and *itpr* RNAi (*1063-R2*) were obtained from National Institute of Genetics Fly Stocks Centre, Kyoto, Japan (NIG). The *UASRNAi* strains for *dSTIM* (47073), *dOrai* (12221), *Arrow* (6707 and 36286), *Dishevelled* (101525), *Shaggy* (101538), *DopECR* (103494), *Dop1R1* (107058), *Dop1R2* (105324), *Dop2R* (11470, 11471), and *TH* (108879) were obtained from Vienna *Drosophila* RNAi center, Vienna, Austria (VDRC). *UASRNAi* strains for Frizzled-2 (BL27568, BL31390 and BL31312) were also obtained from Bloomington Stock Center, Bloomington, IN. RNAi strains are referred to as IR indicating the presence of an interference RNA.

We obtained UAS-DTH1 from Serge Birman (CNRS, ESPCI Paris Tech, France), *UASAcGo* (GoαQ205L) from Yu Fengwei (National University of Singapore, Singapore), *UASAcGi* (GiαQ205L) from Jurgen Knoblich (Institute of Molecular Biotechnology, Austria), *UAS-PTX.16* from Gregg Roman (University of Houston, Texas) and *UASFz2* from Stephen Cohen (Institute of Molecular Cell Biology, Singapore). AcGs (GαsQ215L) BL6490; Go RNAi, BL34653; *UASNaChBac*, BL9468; *UASDishevelled*, BL9453 (*Dsh*); BL9522 (*Dsh$^{G64V}$*), *UASArmadillo$^{active}$*, BL4782 (*Arm$^{act}$*), and *UAS-Shi$^{ts}$* (BL44222) were obtained from the Bloomington Stock Center, Bloomington, IN. *UASdOrai$^+$* (*Venkiteswaran and Hasan, 2009*), *UASdSTIM$^+$* (*Agrawal et al., 2010*), and *UASAcGq3* (*Ratnaparkhi et al., 2002*) have been published. The *GAL80$^{ts}$* strain with two inserts of tubP-GAL*80$^{ts}$* on the second chromosome was generated by Albert Chiang, NCBS, Bangalore, India.

### Flight assay video and electrophysiological recordings

Progeny were collected upon eclosion and aged for 3–4 days. For flight tests, flies were anaesthetized on ice for 15 min and a thin metal wire was glued between the neck and thorax region with the help of

nail polish. To test air-puff-stimulated flight responses, videos were recorded for 30 s after a gentle mouth-blown air puff was delivered to the tethered fly. These videos were analyzed and percentage flight times were calculated. For short flight assays 30 s was taken as 100% flight time. For the long flight assay air-puff-stimulated flight times were monitored for 15 min. For each genotype, a minimum of 30 flies were tethered and tested along with 30 control flies. Flight times of individual flies were noted, and data from a minimum of 30 flies were taken for calculation of the mean and standard error of mean (SEM). Significance testing between the raw data of control and experimental genotypes was performed with the Mann–Whitney U-test using GraphPad Prism 6 (GraphPad Software Inc, La Jolla, CA, USA). Data are represented as bar graphs of the mean percentage flight times. Diamonds inside each bar represent the flight time of individual flies.

Electrophysiological recordings were obtained from the indirect dorsal longitudinal flight muscles (DLMs) as described previously (Banerjee et al., 2004). Briefly, an un-insulated 0.127-mm tungsten electrode, sharpened by electrolysis to attain a 0.5 µm tip diameter, was inserted in the DLMs (fiber a). A similar electrode was inserted in the abdomen for reference. Air-puff stimulated recordings were obtained for 30 s. All recordings were performed using an ISO-DAM8A amplifier (World Precision Instruments, Sarasota, FL) with filter set up of 30 Hz (low pass) to 10 kHz (high pass). Gap free mode of pClamp8 (Molecular Devices, Union City, CA) was used to digitize the data (10 kHz) on a Pentium 5 computer equipped with Digidata 1322A (Molecular Devices). The duration of rhythmic action potential was analyzed using Clampfit (Molecular Devices) and the mean and standard error (SEM) were plotted using Origin 8.0 software (MicroCal, Origin Lab, Northampton, MA, USA). Spike durations in individual flies have been represented as diamonds within the histograms.

## Climbing assay

Progeny were collected upon eclosion and aged for 3–4 days. To test for climbing, flies in batches of 10 were transferred into cylinder of diameter 2.5 cm. Numbers of flies that crossed the 8 cm mark on the cylinder within 12 s, after three gentle taps, were recorded. This procedure was repeated three times with three independent batches of flies. Means and SEM were calculated using the Origin 8.0 software (MicroCal, Origin Lab, Northampton, MA, USA).

## RNA isolation and cDNA synthesis

For isolation of RNA, the central nervous system (CNS) was dissected from adult flies. For each genotype, three independent sets of RNA were isolated each from eight dissected CNS preparations. Total RNA was isolated using TRIzol Reagent (Invitrogen Life Technologies, Carlsbad, CA, USA) according to the manufacturer's specifications. Integrity of RNA was confirmed by visualization on a 1% TAE (40 mM Tris pH 8.2, 40 mM acetate, 1 mM EDTA) agarose gel. Total RNA (500 ng) was treated with DNase in a volume of 45.5 µl with 1 µl (1U) DNase I (Amplification grade, Invitrogen Life Technologies, Carlsbad, CA, USA) with 1 mM dithiothreitol (DTT) (Invitrogen Life Technologies, Carlsbad, CA, USA), 40U of RNase Inhibitor (Promega, Madison, WI, USA) in 5X First Strand Buffer (Invitrogen Life Technologies, Carlsbad, CA, USA) for 30 min at 37°C and heat inactivated for 10 min at 70°C. The reverse transcription reaction was performed in a final volume of 50 µl by addition of 1 µl (200U) Moloney murine leukemia virus (M-MLV) reverse transcriptase (Invitrogen Life Technologies, Carlsbad, CA, USA), 2.5 µl (500 ng) random hexaprimers (MBI Fermentas, Glen Burnie, MD, USA) and 1 µl of a 25 mM dNTP mix (GE Healthcare, Buckinghamshire, UK). Samples were incubated for 10 min at 25°C, then 60 min at 42°C and heat inactivated for 10 min at 70°C. The polymerase chain reactions (PCRs) were performed using 1 µl of cDNA as a template in a 25 µl reaction under appropriate conditions to check the integrity of cDNA prepared.

## Quantitative PCR

Real time quantitative PCR (qPCR) was performed on an ABI 7500 Fast machine (Applied Biosystems, Foster City, California, USA) operated with ABI 7500 software version 2 (Applied Biosystems, Foster City, California, USA) using MESA GREEN qPCR MasterMIx Plus for SYBR Assay I dTTp (Eurogentec, Belgium). Each qPCR experiment was repeated three times with independently isolated RNA samples. qPCRs were performed with *rp49* primers as internal controls and primers specific to gene of interest using dilutions of 1:10. Sequences of the primers used in the 5′ to 3′ directions are given below. The sequence of the forward primer is given first in each case: *dfz2* GGTTACGGAGTGCCAGTCAT; CACAGGAAGAACTTGAGGTCC, *rp49* CGGATCGATATGCTAAGCTGT; GCGCTTGTTCGATCCGTA,

*dsh* CCAAATCCCAAGGGCTACTTC; ATAATACTGTCGTGCGATGTGAG *sgg* GCTGCTCGAGTA TACGCCC; CACTAGGCTGGGCTGTATTGA *th* GTTGCAGCAGCCCAAAAGAAC; GAGACCGTAATC ATTTGCCTTGC.

The cycling parameters were 95℃ for 5 min, 40 cycles of 95℃ for 15 s, and 60℃ for 1 min followed by 1 cycle of 72℃ for 5 min. The fluorescent signal produced from the amplicon was acquired at the end of each polymerization step at 60℃. A melt curve was performed after the assay to check for specificity of the reaction. The fold change of gene expression in the genotype relative to wild-type was determined by the comparative $\Delta\Delta Ct$ method (*Lorentzos et al., 2003*). In this method, the fold change = $2^{-\Delta\Delta Ct}$ where $\Delta\Delta Ct = (C_{t(target\ gene)} - C_{t(rp49)})_{mutant2} - (C_{t(target\ gene)} - C_{t(rp49)})_{Wild\ type}$.

### Immunohistochemistry

Immunohistochemistry was performed on *Drosophila* adult brains expressing cytosolic GFP (*UASGFP*) with the specified *GAL4* strains, after fixing the dissected tissue in 4% paraformaldehyde. The following primary antibodies were used: mouse monoclonal anti-TH antibody (1:50, #22941, ImmunoStar, Hudson, WI, USA), rabbit anti-GFP antibody (1:10,000; #A6455, Molecular Probes, Eugene, OR, USA), mouse anti-Fz2 (1:20; #12A7, DSHB, University of Iowa). 12A7 was deposited to the DSHB by Nusse, R (DSHB Hybridoma Product 12A7). Fluorescent secondary antibodies were used at a dilution of 1:400 as follows: anti-rabbit Alexa Fluor 488 (#A1108) and anti-mouse Alexa Fluor 568 (#A1104, Molecular Probes, Eugene, OR, USA). After antibody staining, confocal analysis was performed on an Olympus Confocal FV1000 microscope and visualized using the FV10-ASW 1.3 viewer (Olympus Corporation, Tokyo, Japan).

### Data analysis

Mean intensity of TH or GFP immunostaining was calculated using ImageJ Version 10.2 (U. S. National Institutes of Health, Bethesda, Maryland, USA, http://imagej.nih.gov/ij/, 1997–2014). Region of interest was drawn around each neuron and mean intensities were obtained for TH and GFP for all the neurons. Median shown by the horizontal line and spread of 25–75% of cell intensities represented as big square was calculated and plotted using Origin 8.0 software (MicroCal, Origin Lab, Northampton, MA, USA) with the data from all the neurons. Significant difference between the different groups of cell intensities was calculated using One-way analysis of variance (ANOVA) for $p < 0.05$.

## Acknowledgements

We thank Dr Krishnamurthy and NCBS Central Image and Flow Facility for help with the confocal imaging. Stocks obtained from the Bloomington Drosophila Stock Center (NIH P40OD018537) were used in this study.

## Additional information

### Funding

| Funder | Grant reference | Author |
| --- | --- | --- |
| National Centre for Biological Sciences (NCBS) | Core funding | Tarjani Agrawal, Gaiti Hasan |
| National Institutes of Health (NIH) | Bloomington *Drosophila* Stock Center P40OD018537 | Gaiti Hasan |
| Tata Institute of Fundamental Research | Core funding | Tarjani Agrawal, Gaiti Hasan |

The funders had no role in study design, data collection and interpretation, or the decision to submit the work for publication.

### Author contributions

TA, Conception and design, Acquisition of data, Analysis and interpretation of data, Drafting or revising the article; GH, Conception and design, Analysis and interpretation of data, Drafting or revising the article

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
