## [Decision Letter]

Thank you for sending your work entitled "Maturation of a central brain flight circuit in *Drosophila* requires Fz2/Ca^2+^ signaling" for consideration at *eLife*. Your article has been favorably evaluated by Eve Marder (Senior editor) and three reviewers, one of whom, Leslie Griffith, is a member of our Board of Reviewing Editors.

The Reviewing editor and the other reviewers discussed their comments before we reached this decision, and the Reviewing editor has assembled the following comments to help you prepare a revised submission.

This paper reports several significant findings that enhance current knowledge about neuronal maturation and flight control in the *Drosophila* brain. The study demonstrates that *dFz2* regulates maturation of a subpopulation of DA neurons called PAM neurons and that these neurons are critical for maintenance of flight. The story is well-developed and interesting, suggesting a very novel role for DA in the MB. There were, however, several concerns shared by the reviewers that should be addressed before the paper is acceptable for publication.

1) The authors use parametric statistical tests for almost all their analysis and it is quite obvious that the data sets are not normally distributed. The analysis has to be done using a non-parametric test or by some mathematical transformation of the data to normalize it before using a parametric test.

2) Figure 7 compares *c305-GAL4* to *MB247* in an attempt to implicate prime lobes in the flight behavior. *c305* is very dirty and has high expression in ellipsoid body, SEG and AL. Authors need to r/o EB and other areas as the locus of DAR action: either show that *MB-GAL80* blocks their phenotype or that a cleaner prime lobe line (Janelia has many) gives the same phenotype.

3) The conclusions regarding *dFz2* are based on a single RNAi line. Did the authors try any other *dFz2*-RNAi lines with the *THGAL4,GAL80*^*ts*^ driver? An independent line would add to the confidence in the result.

---

## [Author Response]

*1) The authors use parametric statistical tests for almost all their analysis and it is quite obvious that the data sets are not normally distributed. The analysis has to be done using a non-parametric test or by some mathematical transformation of the data to normalize it before using a parametric test*.

We have re-done the statistical analyses for all datasets that are not normally distributed using the Non-parametric Mann–Whitney U-test. This has been mentioned in the Methods section in the subsection headed “Flight Assay Video and Electrophysiological Recordings” and in all relevant figure legends.

*2)*
Figure 7
*compares* c305-GAL4 *to* MB247 *in an attempt to implicate prime lobes in the flight behavior.* c305 *is very dirty and has high expression in ellipsoid body, SEG and AL. Authors need to r/o EB and other areas as the locus of DAR action: either show that* MB-GAL80 *blocks their phenotype or that a cleaner prime lobe line (Janelia has many) gives the same phenotype*.

We obtained a splitGAL4 line from the Janelia collection that specifically and uniquely marks the prime lobes of the MB (*MB186BGAL4*; Aso et al., *eLife*, 2014). Flies with *MB186B* driven *Shi*^*ts*^ expression and *Dop1R2* knockdown exhibit significant flight deficits (Figure 7 and Figure 7). Data from the earlier version of the manuscript with *c305aGAL4* is now shown in Figure 7—figure supplement 2. The Results have been modified accordingly (please see the subsection entitled “Maintenance of acute flight requires synaptic activity in α’ β’ lobes of the mushroom body”).

*3) The conclusions regarding* dFz2 *are based on a single RNAi line. Did the authors try any other* dFz2*-RNAi lines with the* THGAL4,GAL80 ^ts^
*driver? An independent line would add to the confidence in the result*.

Three additional RNAi lines from Bloomington (*BL27568*, *BL31390*, *BL31312*) were tested with *THGAL4;GAL80*
^*ts*^ (Figure 1—figure supplement 2). All three exhibit flight phenotypes but none of these resulted in as strong a phenotype as shown by the original NIG line (*9739R-1(II)* referred to as *dFz2-IR*; Figure 1—figure supplement 2). We tested the knockdown efficacy of one amongst the three BL lines (*BL27568*) considered most effective in somatic tissues as it is was generated with the VALIUM10 vector. As shown in Figure 1—figure supplement 2, knockdown of *dFz2* RNA by *BL27568* was mild, thus supporting the flight data obtained with this line (Figure 1—figure supplement 2).